

# Integrative analyses of single-cell and bulk RNA sequencing to construct the tumor-associated macrophage-related prognostic signature in lung adenocarcinoma

Chaoqun Yu[1,*], Jun Chen[2,*], Jianming Deng[3], Hui Li[4], Qianru Zhuang[5], Bingqing Luo[6], Hua Ye[7] and Hui Tian[1]

[1] Thoracic Surgery, Ningbo Medical Centre Lihuili Hospital, Ningbo, China
[2] Department of Oncology, The First People's Hospital of Zhaoqing, Zhaoqing, China
[3] Thoracic Surgery, Guangdong Second Provincial General Hospital, Guangzhou, China
[4] Department of Thoracic Surgical Ward II, Shandong Cancer Hospital and Institute, Shandong First Medical University and Shandong Academy of Medical Science, Jinan, Shandong, China
[5] Oncology Department 2, Maoming People's Hospital, Maoming, China
[6] Department of Respiratory Oncology, The Second Affiliated Hospital of Xiamen Medical College, Xiamen, China
[7] Department of Respiratory and Critical Care Medicine, Yueqing People's Hospital, Yueqing, Zhejiang, China
* These authors contributed equally to this work.

Corresponding authors
Hua Ye, yhdoctor@126.com
Hui Tian, tianhdoctor@163.com

## ABSTRACT

**Background**. Tumor-associated macrophages (TAMs) are a type of tumor-infiltrating immune cell that play a crucial role in tumor progression. However, the roles of TAMs and their regulatory mechanisms in lung adenocarcinoma (LUAD) remain poorly understood. Therefore, we aimed to develop a novel TAM-related prognostic signature to predict survival outcomes and constructed a lncRNA-miRNA-mRNA network based on these genes.

**Methods**. Transcriptomic data, clinical data, and single-cell RNA-sequencing (scRNA-seq) data were obtained from LUAD patients were obtained from the Cancer Genome Atlas (TCGA) and Gene Expression Omnibus (GEO) databases. Differentially expressed-lncRNAs (DE-lncRNAs), miRNAs (DE-miRNAs), and mRNAs (DEGs) were identified in LUAD. Differentially expressed TAM-related genes (DE-TAMGs) were selected and used to construct prognostic signatures. A TAM-related risk score was calculated, and patients were stratified into high- and low-risk groups based on the median risk score. Then, biological functions, immune characteristics, and responses to immunotherapy and chemotherapy were assessed across the risk groups. A TAM-related lncRNA-miRNA-mRNA network was constructed based on DE-lncRNAs, DE-miRNAs, and TAM-related signatures. Quantitative polymerase chain reaction (qPCR) was used to validate the expression of TAM-related genes, and scRNA-seq analysis was used to examine cell-type-specific expression of these genes.

**Results**. A total of 316 DE-lncRNAs, 162 DE-miRNAs, and 2,601 DEGs were screened in LUAD. Among these, 147 DE-TAMGs were selected. KLF4, GAPDH PDGFB, TIMP1, CD74, and CCL20 were identified as the key prognostic markers in LUAD. Patients were divided into high- and low-risk groups based on the median risk score.
Enrichment analysis revealed several cancer-related pathways associated with the high-risk group, and significant differences in terms of immune cell infiltration, HLA-related gene expression, immune checkpoints expression, and therapeutic responses were observed between high- and low-risk groups. We also constructed a lncRNA-miRNA-mRNA network, which included 36 DE-lncRNAs, 23 shared miRNA, and four TAMGs (PDGFB, CD74, KLF4, and CCL20). The qPCR results indicated the increased expression of PDGFB, CD74, and KLF4 but decreased expression of CCL20 in LUAD tumor tissues compared with adjacent normal tissues. scRNA-seq analysis revealed that CD74, KLF4, CCL20, PDGFB were specifically expressed in macrophages.

**Conclusions**. In conclusion, we identify the TAM-related prognostic signature that predicts the survival outcome in patients with LUAD. This signature may offer a novel effective therapeutic strategy for LUAD patients.

## INTRODUCTION

Lung cancer remains the second most commonly diagnosed cancer and the leading cause of cancer-related death worldwide, with 2.2 million new cancer cases and 1.8 million deaths in 2020 (*Sung et al., 2021*). Non-small cell lung carcinoma (NSCLC) is the most common subtype of lung cancer, consisting of lung adenocarcinoma (LUAD), squamous cell carcinoma (SCC), and large cell carcinoma (*Herbst, Heymach & Lippman, 2008*). Over the past two decades, advancements in targeted therapy and immunotherapy have led to declines in lung cancer incidence (*Jasper et al., 2022*; *Mithoowani & Febbraro, 2022*). However, survival rates for patients with advanced NSCLC remain lower (*Siegel et al., 2022*; *Siegel et al., 2023*). Therefore, understanding of genetic alterations driving NSCLC initiation and progression is important.

Lung cancer is a highly heterogeneous disease that closely links to the complex tumor microenvironment (TME) (*Altorki et al., 2019*). The TME consists of cancer cells, infiltrating immune cells, stromal cells, and other cell types together with noncellular tissue components, all of which play an crucial roles in tumorigenesis, development, aggression, metastasis, and drug resistance (*De Visser & Joyce, 2023*; *Xiao & Yu, 2021*). Tumor-associated macrophages (TAM) are a major population of tumor-infiltrating immune cells, exhibiting dual-roles in tumor progression with two major phenotypes, M1, the tumor-suppressing subtype, M2, the tumor-promoting type (*Franklin et al., 2014*; *Mantovani et al., 2017*; *Murray et al., 2014*). Recent studies have shown that TAM can remodel the tumor immune microenvironment, further influencing responses to targeted therapies and immunotherapies (*Boutilier & Elsawa, 2021*; *Dai et al., 2020*; *Kumari & Choi, 2022*; *Pan et al., 2020b*). Therefore, exploring novel regulatory mechanisms of TAMs that are linked to tumor initiation, development, and prognosis is essential.

Long non-coding RNAs (lncRNAs) are a class of non-coding RNAs (ncRNAs) longer than 200 nucleotides (nt) that have little or no protein-coding capacity (*Osielska &*
*Jagodziński, 2018*; *Yan & Bu, 2021*). Typically, lncRNAs act as competitive endogenous RNAs (ceRNAs), binding to micro RNAs (miRNAs) and competing for their target genes (*Wang et al., 2019*). Emerging evidence has indicated that lncRNAs play crucial roles in tumorigenesis and metastasis (*Cheng et al., 2022*; *Hua et al., 2019*; *Lu et al., 2017*). Specifically, M2 macrophage-derived exosomal lncRNAs contribute to resistance to radiotherapy in NSCLC cells (*Zhang et al., 2021*). Additionally, LINC00273 and LINC00963 mediate the communication between LUAD cells and TAMs, promoting tumorigenesis and progression (*Chen et al., 2021*; *Wang et al., 2023*), while lncRNA SNHG7 mediates M2 polarization in macrophages, enhancing docetaxel resistance in LUAD (*Zhang et al., 2022*). Abundant evidence revealed that lncRNAs play important roles in lung cancer, particularly in influencing the fate of TAMs in TME.

In the present study, we identified the TAM-related genes (TAMGs), lncRNAs, miRNAs, and their regulated network associated with the prognosis of LUAD. We also explored their prognostic impact, effects on tumor immune microenvironment, and therapeutic sensitivity in LUAD.

## MATERIALS AND METHODS

### Data collection and processing

The RNA-seq data and miRNA-seq data of LUAD, along with the corresponding clinical data, were downloaded from The Cancer Genome Atlas (TCGA, https://www.cancer.gov/ccg/research/genome-sequencing/tcga). After excluding the sample with incomplete follow-up information, a total of 486 LUAD and 59 normal samples from the RNA-seq data, as well as a total of 521 LUAD and 46 normal samples the miRNA-seq data, were included for subsequent analysis in this study. The mRNAs and lncRNAs of samples were annotated based on the gene transfer format (GTF) files containing the gene symbol. Additionally, transcriptomic profiles and corresponding clinical information in the GSE31210 dataset, which includes 226 tumor samples, were obtained from Gene Expression Omnibus (GEO, https://www.ncbi.nlm.nih.gov/geo/) and generated by GPL570 platforms. Moreover, scRNA-seq files from nine LUAD tumor specimens were obtained from the GEO database (accession number GSE171145), generated by the GPL24676 platform (Illumina NovaSeq 6000). Finally, a total of 428 tumor-associated macrophage (TAM)-related genes with a score greater than five were obtained from the GeneCards database using keywords searches (Table S1).

### Screening differentially expressed RNAs in LUAD

Limma R package was performed to identify the differentially expressed lncRNAs (DE-lncRNAs), mRNAs (DEGs), and miRNAs (DE-miRNAs) according to the criteria of |log2 (fold change, FC)|> 1 and adjusted $P$ value < 0.05. Volcano plots were drawn using the ggplot2 R package to visualize the differentially expressed RNAs, and a heatmap was drawn using the pheatmap R package to illustrate the top50 DE-lncRNAs, top50 DEGs, and all DE-miRNAs.

## Construction and validation of a TAM-related gene signature

The differentially expressed TAMGs (DE-TAMGs) were identified by overlapping the DEGs and 428 TAMGs. The 486 patients from the TCGA-LUAD cohort were randomly split into training and testing sets at a 7:3 ratio using the caret R package. Afterward, the DE-TAMGs that were associated with survival were identified using univariate Cox analysis, these DE-TAMGs were incorporated into a least absolute shrinkage and selection operator (LASSO) regression model to select the prognostic gene signature by glmnet R package. A risk score was calculated as follows, risk score $= \sum_1^i (CoefGenei * ExpGenei)$. Coef represents the regression coefficient, and ExpGene represents the expression values of the gene. All patients were stratified into high- and low-risk groups based on the median risk score. The correlation between risk score and clinical characteristics (age, gender, pathologic stage, and T/N/M stages) were analyzed using Pearson correlation analysis. The Kaplan–Meier method and log-rank test were applied to compare the overall survival (OS) of patients between high- and low-risk groups using the survival R package. The time-dependent receiver operator characteristic (ROC) curve was conducted to evaluate the predictive accuracy of the risk model using the survivalROC R package.

## Gene Set Enrichment Analysis (GSEA)

The pathway enrichment between high- and low-risk groups was analyzed using GSEA. The c2.cp.kegg.v7.4.symbols.gmt and h.all.v7.4.symbols.gmt were selected from the Molecular Signatures Database (MSigDB, https://www.gsea-msigdb.org/gsea/msigdb) for the enrichment analysis, which was performed using the GSEA R package.

## Tumor immune microenvironment analysis

The ESTIMATE algorithm was used to evaluate the immune score, stromal score, and ESTIMATE score using the estimate R package. Differences in immune score, stromal score, and ESTIMATE score between high- and low-risk groups were detected using the Wilcoxon-test. Besides, the CIBERSORT algorithm was performed to evaluate the abundance of 22 types of immune cells based on the gene signatures consisting of 547 genes. The significant differences in infiltrated immune cells between high- and low-risk groups were detected using the Wilcoxon test.

## Immunotherapeutic and chemotherapeutic effect analysis

Human leukocyte antigen (HLA) expression plays a key role in tumor immunogenicity and responses to immunotherapies (*Gong & Karchin, 2022*). Thus, the differences in HLA genes and immune checkpoint levels between high- and low-risk groups were detected using Wilcoxon test. The Tumor Immune Dysfunction and Exclusion (TIDE) (*Lu et al., 2019*), Immunophenoscore (IPS) (*Charoentong et al., 2017*), and the submap algorithm were used to predict the response to immune checkpoint blockade. A *P* value < 0.05 was considered statistically significant. The pRRophetic algorithm was conducted to estimate the therapeutic response of the samples with different risk scores based on the Genomics of Drug Sensitivity in Cancer (GDSC).

## Construction of a lncRNA-miRNA-mRNA network

The miRNA-mRNA pairs were identified based on risk signatures and DE-miRNAs in the online databases, including miRcode (https://bio.tools/miRcode), miRDB (https://mirdb.org/), miRTarBase (https://mirtarbase.cuhk.edu.cn/), Starbase (https://starbase.sysu.edu.cn/), and TargetScan (https://www.targetscan.org/). Additionally, miRNA-lncRNA pairs were screened out based on DE-lncRNA in Starbase (https://starbase.sysu.edu.cn/). Finally, an endogenous competitive network, lncRNA-miRNA-mRNA, was constructed based on the DE-lncRNAs, shared-miRNA that lncRNA-miRNA and miRNA-mRNA pairs, and risk gene signature. The network was visualized using Cytoscape Version 3.8.0.

## Clinical specimen

Tumor tissues and paracancerous non-tumor tissue samples were collected from lung cancer patients ($n = 28$) at Yueqing People's Hospital. Histological examination was used to confirm the diagnosis of LUAD. All procedures were approved by the Ethical Committee of Yueqing People's Hospital (YQYY202300221). Written informed consent was obtained from all participants prior to their inclusion in the study. The tissue specimens were immediately frozen in liquid nitrogen after collection and stored at $-80\ °C$ for further RNA extraction.

## Experimental validation of the TAM-related genes involved in ceRNAs

Total RNA was extracted and purified using TRIzol (TAKARA, Dalian, China) according to the manufacturer's instructions. cDNA synthesis was performed using a Bestar qPCR RT Kit (DBI Bioscience, Shanghai, China) according to the manufacturer's protocol. Data were collected as previously described in *He et al. (2016)*, Specifically the experimental validation of the TAM-related genes involved in ceRNAs. Quantitative real-time PCR (RT-qPCR) analyses were performed with a Bestar® SybGreen qPCR Mastermix (DBI Bioscience, Shanghai, China) following: 95 °C for 20 s, followed by 40 cycles of 95 °C for 1 s and 60 °C for 20 s. GAPDH was used as the reference gene for the normalization of all gene expression results. The average of three independent analyses for each gene was calculated. The fold changes were calculated through relative quantification ($2^{-\Delta\Delta Ct}$). All reactions were run in triplicate and repeated in three independent experiments. The primers used were as follows: GAPDH forward, 5′-TGTTCGTCATGGGTGTGAAC-3′ and reverse 5′-ATGGCATGGACTGTGGTCAT-3′. PDGFB forward, 5′-TTATCATGGGCCTCGGGGA-3′ and reverse 5′-CAGACGGACGAGGGAAACAA-3′. CD74 forward, 5′-GGCTACTGCTGGTGTGTCTT and reverse,5′-TCCAAGGGTGACGAAAGAGC-3′. KLF4 forward, 5′-GTCCCGGGGATTTGTAGCTC-3′ and reverse 5′-TGTAGTGCTTTCTGGCTGGG-3′. CCL20 forward, 5′- TGTCAGTGCTGCTACTCCAC-3′ and reverse 5′-ACAAGTCCAGTGAGGCACAA-3′.

## Data processing and single cell analysis

The Serut R package was used for standard downstream processing for scRNA-seq data. Gene expression was required in at least three cells, with a gene count more than 300 and less than 7,000, the mitochondria proportion less than 10%, and the erythrocyte proportion less than 3%. Cells that did not meet these criteria were excluded from the

analysis. Afterward, "NormalizeData" was used to perform the data normalization. PCA (Principal Component Analysis) was used to reduce the dimensionality of the data, after which gene expression value were converted to z-score using "ScaleData" function. Then, Uniform Manifold Approximation and Projection for Dimension Reduction (UMAP) was utilized for unsupervised clustering and unbiased visualization of cell populations on a two-dimensional map. To account for cell cycle differences, "CellCycleScoring" was used to detect cell cycle-related genes. "FindAllMarkers" was performed to identify marker genes of each cluster with the threshold value of |log (fold change, FC)| > 0.025 and a minimum cell population fraction of 0.25 in either of the two populations. The SingleR package was utilized for cell-type annotation. The classical gene markers of LUAD were visualized as bubble plots by 'DotPlot'. The hub gene expression across different cell types, along with significant cell types, was selected for biological function (GO and KEGG pathway enrichment) analysis using clusterProfiler R package. Monocle2 R package was used for cell trajectory and pseudo-time analysis with the DDRTree method.

## RESULTS

### Identification of the differential expression of RNAs in LUAD

The flowchart of this study are shown in Fig. S1. In the present study, the limma R package was performed to identify the differential expression of RNAs in LUAD, resulting in a total of 316 DE-lncRNAs (116 upregulated and 200 downregulated DE-lncRNAs, Figs. 1A, 1D), 162 DE-miRNAs (125 upregulated and 37 downregulated DE-miRNAs, Figs. 1B, 1E), and 2,601 DEGs (924 upregulated and 1,677 downregulated DEGs, Figs. 1C, 1F) were identified in LUAD (Tables S2–S4).

### Construction of a TAM-related gene signature in LUAD

A total of 147 DE-TAMGs were identified from the 2,601 DEGs (Fig. 2A) and subsequently incorporated into a univariate Cox regression model, which revealed 12 DE-TAMGs with significant associations ($p$-value < 0.05) in the training set, including CAT, KLF4, CLEC12A, GAPDH, PDGFB, HSPD1, TIMP1, ITGAL, CA9, TNFSF11, CD74, CX3CL1, FOLR1, CLEC10A, and CCL20 (Fig. 2B). LASSO regression analysis further narrowed down the candidate genes to KLF4, GAPDH PDGFB, TIMP1, CD74, and CCL20, which were ultimately selected to form the prognostic signature for LUAD (Figs. 2C–2D, Table S5).

The risk score for each patient was calculated based on the expression levels and corresponding regression coefficients of these six genes. Patients in the training set were subsequently divided into high-risk ($n = 179$) and low-risk ($n = 163$) groups (Table S6). We found that risk scores were associated with the ages and gender of patients (Figs. 2E–2G). Additionally, a higher risk score correlated with tumorigenesis, lymph node metastasis, and distant metastasis, and more aggressive progression (Figs. 2E, 2H–2K).

### Validation of the TAM-related gene signature in LUAD

To validate the prognostic potential of the TAMG-related risk score, we applied it to several datasets, including the training TCGA-LUAD set, the test TCGA-LUAD set, the entire TCGA-LUAD set, and the external GSE31210 set. Based on the median risk score,

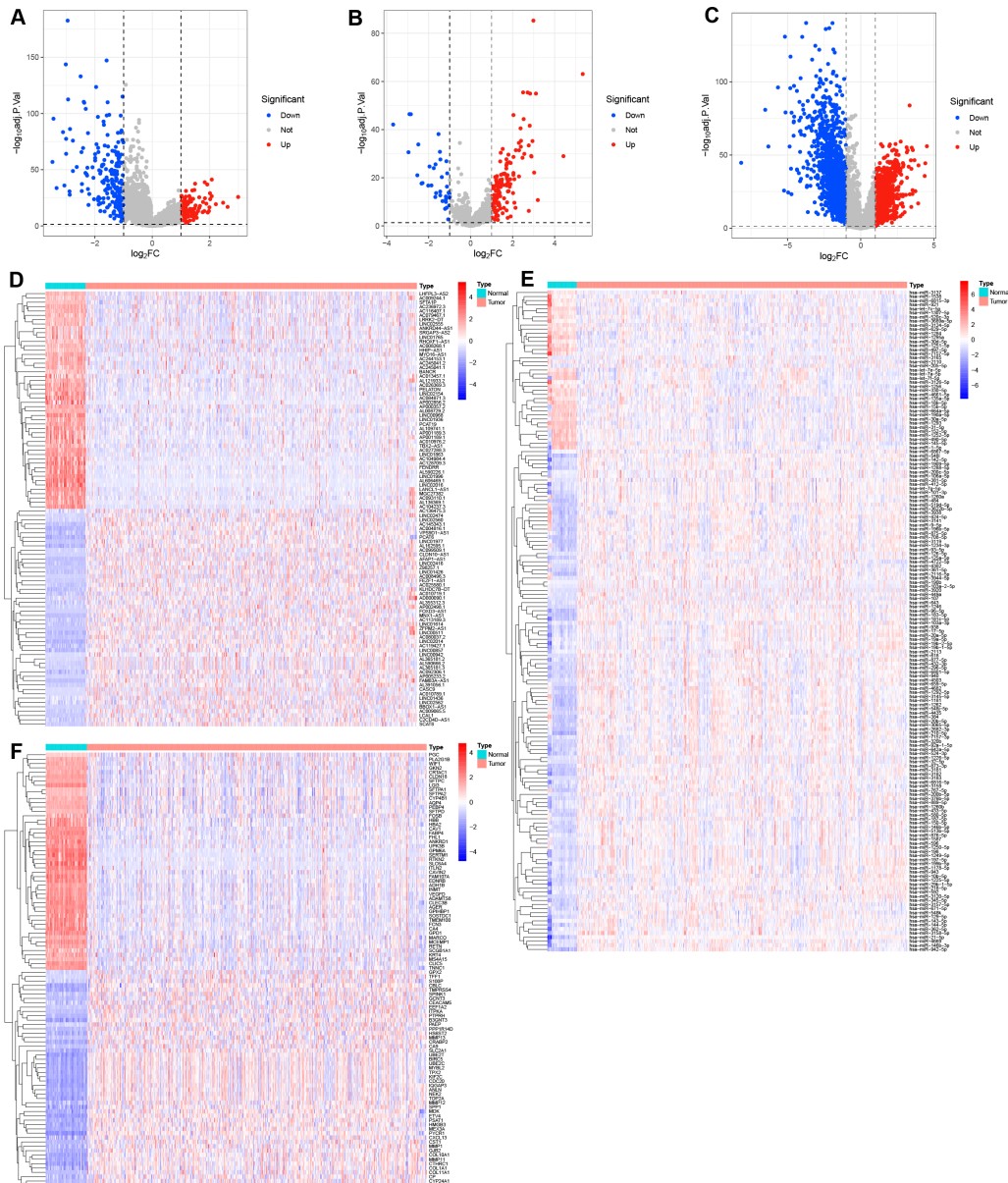

**Figure 1** **Identification of the differential expression of RNAs in LUAD.** (A–C) Volcano plots the DE-lncRNAs, DE-miRNAs, and DEGs between LUAD and normal tissues with the criteria of |log2 FC| > 1 and adjusted *P* value < 0.05. (D–F) Heatmaps of the top 50 DE-lncRNAs, all DE-miRNAs, and top 50 DEGs in LUAD.

patients were categorized into high- and low-risk groups. Patients in the high-risk group showed worse survival status compared to those in the low-risk group (Figs. 3A–3D). Kaplan–Meier OS survival curves indicated significantly poorer prognoses for patients in the high-risk group compared to those in the low-risk group (Figs. 3E–3H). Time-dependent ROC curves for those sets at 1-, 2-, 3-, 4-, and 5 years showed the risk model

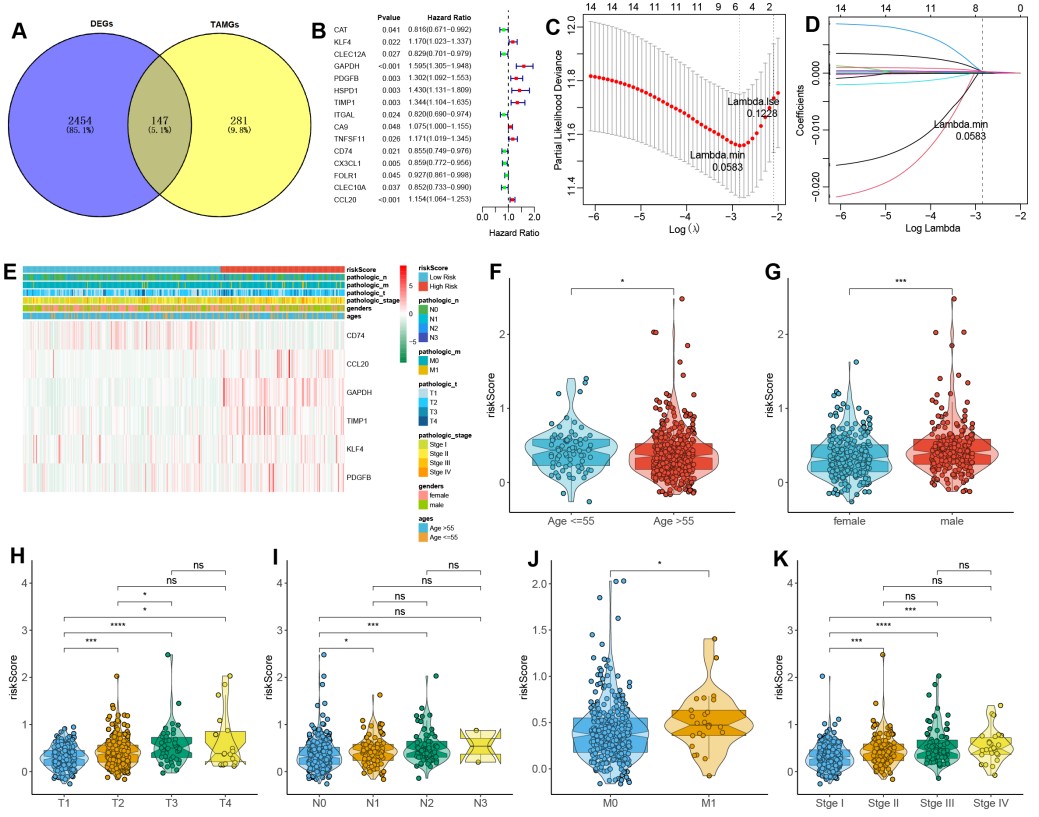

**Figure 2** **Construction of a TAM-related gene signature in LUAD.** (A) Venn plot of the DE-TAMGs by overlapping DEGs and TAMGs from GeneCards. (B) Univariate Cox regression analysis to investigate the DE-TAMGs linked to survival. (C) The distribution plot of the partial likelihood deviation of the LASSO coefficient. (D) The distribution plot of the LASSO coefficient. (E) Heatmap of the correlation between risk score and clinical characteristics (age, gender, pathological stages, and T/N/M stages). (F–K) Violin plots of the correlation between risk score and clinical characteristics (age, gender, pathological stages, and T/N/M stages).

and TAM-related gene signature could predict the OS of LUAD patients with moderate sensitivity and specificity (Figs. 3I–3L).

## TAM-related functional enrichment analysis

Based on previous analyses, we further conducted the enrichment analysis using GSEA to explore the functional pathways associated with the TAM-related risk score. The results indicated that several canonical cancer pathways were significantly enriched between the high- and low-risk groups (Figs. 4A–4B). Specifically, we found that the cell cycle, NOTCH signaling pathway, p53 signaling pathway, pathways in cancer, proteasome, hypoxia, PI3K/AKT/mTOR signaling pathway, reactive oxygen species pathway, and Wnt/beta-catenin signaling pathway were significantly enriched in the high-risk group compared to the low-risk group (Figs. 4A–4B).

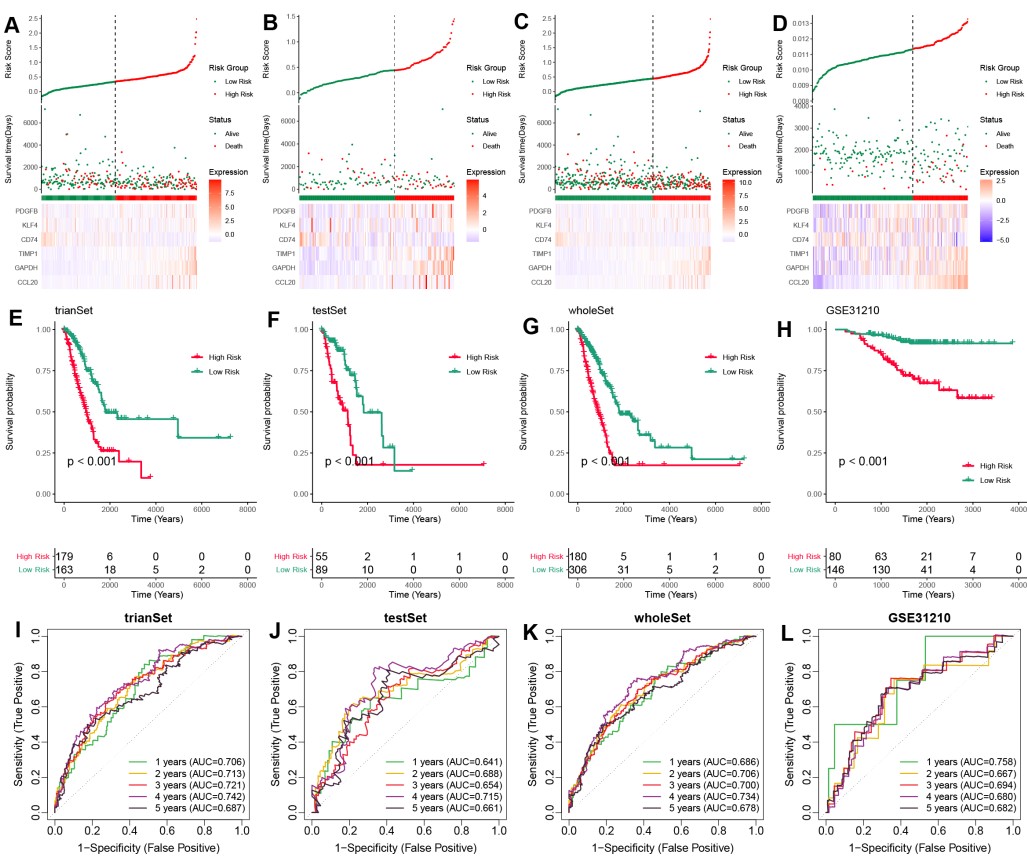

**Figure 3** **Validation of the TAM-related gene signature in LUAD.** (A–D) The distribution of risk scores based on the TAM-related signature in the training-TCGA, test-TCGA, entire-TCGA, and GSE31210 datasets. Top: Patients were divided into high- and low-risk groups based on the median value of the risk score. Medium: The distribution of patients with survival status. Bottom: The heatmap of the differential expression of the TAM-related signature between high- and low-risk groups. (E–H) Kaplan–Meier analysis of the OS of high- and low-risk groups in training-TCGA, test-TCGA, entire-TCGA, and GSE31210 datasets. (I–L) Time-dependent ROC curves of the 1-, 3-, 5-years OS of high- and low-risk groups in the training-TCGA, test-TCGA, entire-TCGA, and GSE31210 datasets.

## Correlation analysis between TAM risk score and tumor immune microenvironment

We further investigated the immune characteristics within the tumor microenvironment and observed that the low-risk group had higher immune and ESTIMATE scores compared to the high-risk group (Figs. 5A–5C). In addition, we found significant differences s in the proportions of various immune cell types (Fig. 5D). In the high-risk group, the proportions of macrophages M0, T cells CD4 memory activated, neutrophils, NK cells resting, Mast cells activated was upregulated, but the fraction of T cells CD4 memory resting, mast cells resting, dendritic cells resting, T cells regulatory (Tregs), monocytes, and B cells memory were downregulated compared to the low-risk group (Fig. 5E). Macrophages M0 are unactivated macrophages, typically considered to be in a "resting" or undifferentiated state. They perform various immune surveillance and regulatory functions but usually require

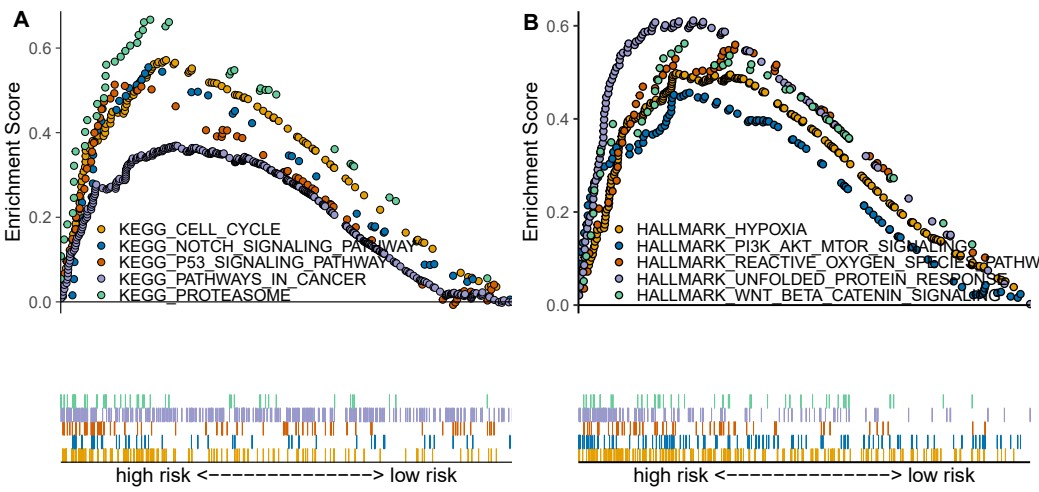

**Figure 4** **TAM-related functional enrichment analysis.** (A) KEGG pathways analysis between high- and low-risk groups. (B) Hallmark pathways analysis between high- and low-risk groups.

specific immune signals to differentiate into more immunologically active M1 or M2 macrophages. In the analysis of TAM-related high- and low-risk groups, M0 macrophages were significantly increased in the high-risk group, indicating a close association between TAMs and high-risk status.

## Correlation analysis between TAM risk score and immunotherapeutic effect

We next compared the expression of HLA-related genes and immune checkpoint markers between high- and low-risk groups. Most HLA-related genes, including HLA class I (HLA-A/B/C/E/F) and HLA class II (HLA-DMA/B, HLA-DOA/B, HLA-DPA1/B1, HLA-DPQ1/B1, HLA-DRA1/B1, and HLA-DRB5), were downregulated in high-risk group than low-risk group (Fig. 6A). Similarly, the expression of immune checkpoint genes, such as CD80, CTLA4, and TIGIT was downregulated in a high-risk group than the low-risk group (Fig. 6B).

As expected, the TIDE score was significantly higher in a high-risk group than in a low-risk group (Fig. 6C), with a strongly positive correlation between risk score and TIDE score (Fig. 6D). In contrast, the IPS score was lower in the high-risk group compared to the low-risk group (Fig. 6E). These findings suggest that the low-risk group exhibits more active immunological functions and greater sensitivity to immunotherapy. However, the patients in the low-risk group did not show significant responses to anti-PD1 and anti-CTLA4 therapies (Fig. 6F).

## Correlation analysis between TAM risk score and chemotherapeutic effect

We also identified the potential chemotherapeutic agents targeting the TAM-related prognostic signature. A total of 13 compounds showed significantly different half-inhibitory
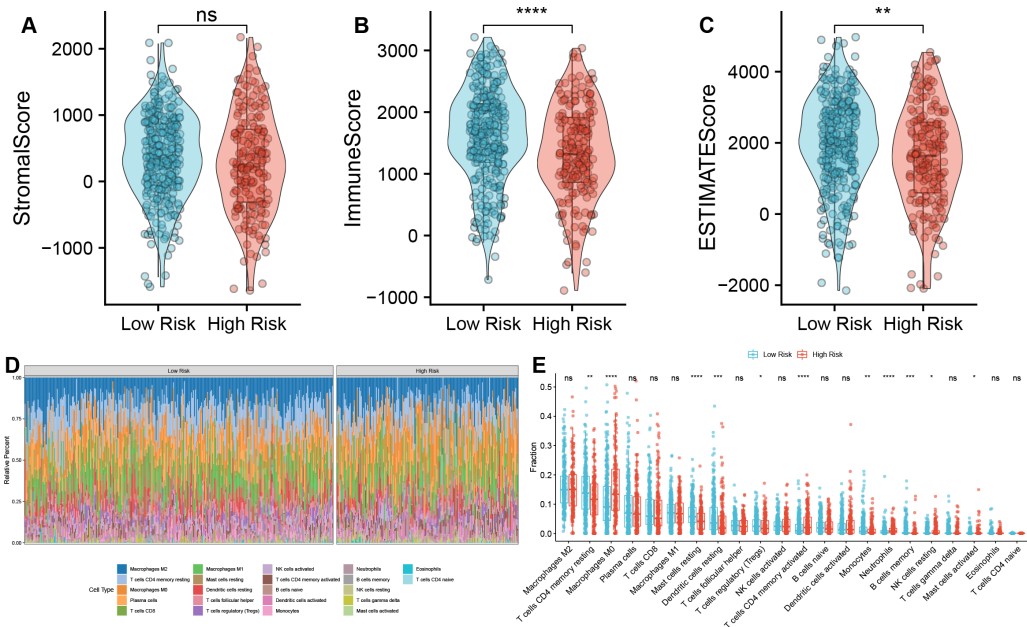

**Figure 5** Correlation analysis between TAM risk score and tumor immune microenvironment. (A–C) Violin plots of the differences in Strome score, immune score, and ESTIMATE score between high- and low-risk groups. (D) Heatmap of the fractions of infiltrated immune cells between high- and low-risk groups. (E) The histogram of the different infiltrated immune cells between high- and low-risk groups.

concentrations (IC50) between the high- and low-risk groups (Table S7). Most of these compounds exhibited higher IC50 values in the high-risk group than low-risk group (Fig. 7).

## Development of a TAM-associated lncRNA-miRNA-mRNA network in LUAD

Using several online databases, including miRcode, miRDB, miRTarBase, Starbase, and TargetScan, a total of 66,130 miRNA-mRNA pairs were identified (Table S8), From these, 24 miRNA-TAMG (PDGFB, CD74, KLF4, and CCL20) pairs were selected for further analysis (Table S9). In addition, a total of 7,975 miRNA-lncRNA pairs were screened (Table S10).

Subsequently, a lncRNA-miRNA-mRNA network was constructed, incorporating 36 DE-lncRNAs, 23 shared-miRNA, and four TAMGs (PDGFB, CD74, KLF4, and CCL20, Fig. 8A). QPCR results confirmed that CCL20 expression was significantly upregulated in tumor tissues (Fig. 8E), while the expression of PDGFB, CD74, and KLF4 was downregulated in tumor tissues compared to adjacent normal tissues (Figs. 8B–8D). Additionally, a similar trend was observed in the TCGA-LUAD cohort, where CCL20 expression was significantly upregulated in tumor tissues, whereas PDGFB, CD74, and KLF4 were significantly downregulated compared to adjacent normal tissues (Figs. 8F–8I).
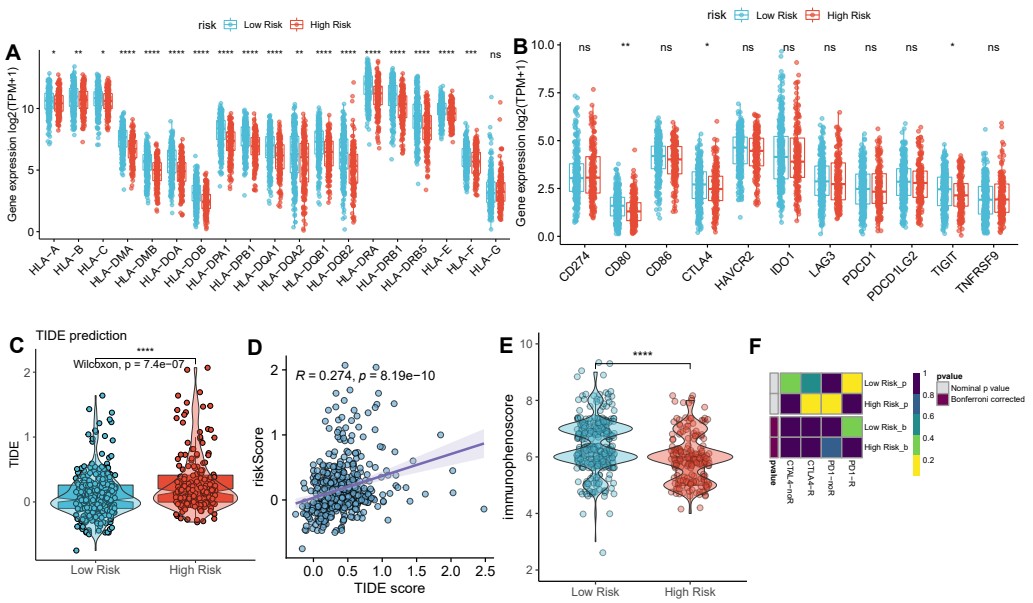

**Figure 6 Correlation analysis between TAM risk score and immunotherapeutic effect.** (A–B) The histograms of the differentially expressed of HLA-related genes and immune checkpoints between high- and low-risk groups. (C) Violin plot of TIDE scores between high- and low-risk groups. (D) Scatter plots indicated the correlation between TIDE score and risk score. (E) Violin plot of IPS scores between high- and low-risk groups. (F) Submap of immunotherapeutic responses by anti-PD1 and anti-CTLA4 treatment.

## Validation of the key TAMG expression in monocyte/macrophage at the single-cell level

After the quality control of scRNA-seq data, a total of 30,874 cells were selected for subsequent analysis (Figs. S2A–S2B). After normalization and dimensionality reduction, 27 subpopulations were obtained (Figs. S2C–S2I). Then, the relative expression of marker genes in each cluster was presented in the heatmap (Fig. S2J)). Afterward, 17 cell types were annotated using a singleR package (Fig. 9A), including CD4 T cells, CD8 T cells, monocytes, B cells, secretory club cells (Club), cancer cells, macrophages, alveolar cells, NK cells, plasma, mast cells, smooth muscle cell, cycling cells, neutrophils, fibroblasts, endothelial cells, and ciliated airway epithelial cells (Ciliated).

The marker genes for each cell type, along with the top three markers, are shown in Figs. 9B–9C. Furthermore, we examined the expression of PDGFB, CD74, KLF4, and CCL20 in different cell types, with a focus on monocytes/macrophages. Our analysis indicated that CD74, KLF4, and CCL20 were significantly expressed in these cells, whereas PDGFB was downregulated (Fig. 9D, Figs. S3A–S3C).

GO enrichment analysis of macrophage marker genes indicated that significant enrichment in protein-macromolecule adaptor activity, nucleoside-triphosphatase regulator activity, GTPase regulator activity, and chaperone binding (Fig. 10A). For monocytes, marker genes were enriched in kinase regulator activity, DNA-binding transcription activator activity, and RNA polymerase II-specific functions (Fig. 10A).

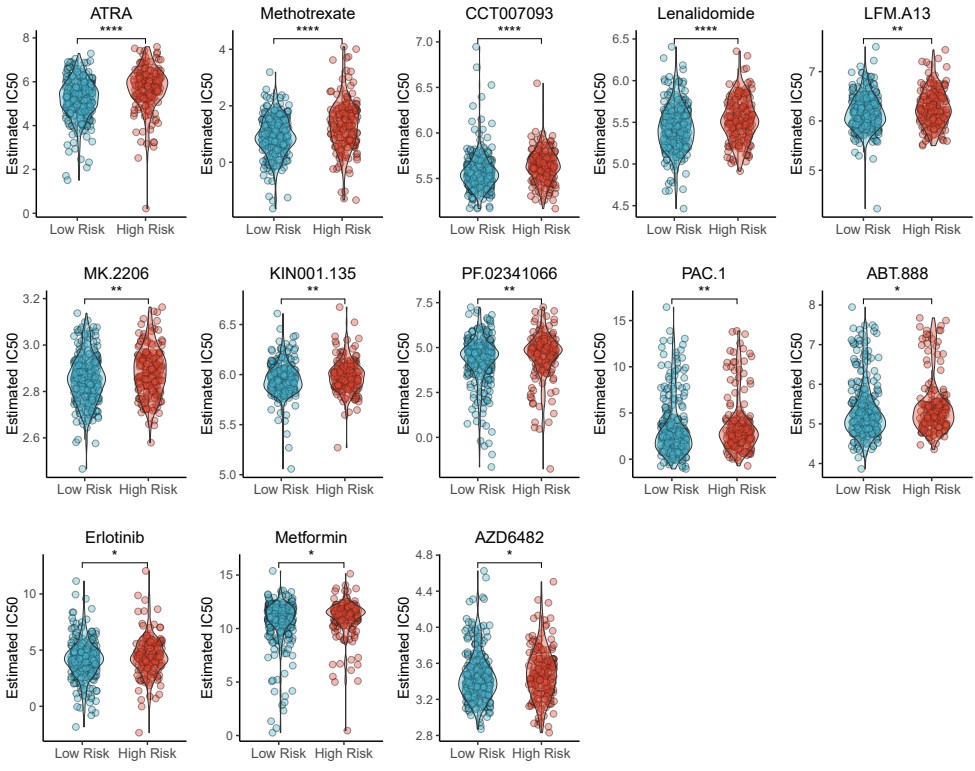

**Figure 7** **Violin plots of correlation of TAM risk score and chemotherapeutic effect.**

KEGG pathway enrichment analysis indicated that the macrophage marker genes were mainly associated with calcium reabsorption regulation, galactose metabolism, and the IL-17 signaling pathway (Fig. 10B). In contrast, monocyte marker genes were significantly associated with serotonergic synapses, the Apelin signaling pathway, and non-small cell lung cancer (Fig. 10B).

Finally, cell trajectory and pseudo-time analysis of monocyte and macrophages using the monocle2 R package revealed a clear transition from monocytes to macrophages (Figs. 10C–10F). These results support the notion that tissue-resident macrophages originate from hematopoietic stem cells and monocyte-derived macrophages (*Lazarov et al., 2023*).

# DISCUSSION

Increasingly studies have shown that TAMs are a major component of TME and are implicated in poor prognosis and therapy resistance across various cancers (*Cassetta & Pollard, 2020*; *Xiang et al., 2021*). Previous research demonstrated that TAMs drive cancer malignancy by mediating angiogenesis, promoting tumor invasion and metastasis, dysregulating metabolism, and promoting tumor hypoxia and immunosuppressive microenvironment (*Cheng et al., 2021*; *Jeong et al., 2019*; *Pu & Ji, 2022*; *Vitale et al., 2019*). Advances in therapeutic strategies have highlighted that targeting TAMs can synergize tumor immunotherapy, thereby improving treatment efficacy (*Bai et al., 2022*; *Binnewies*
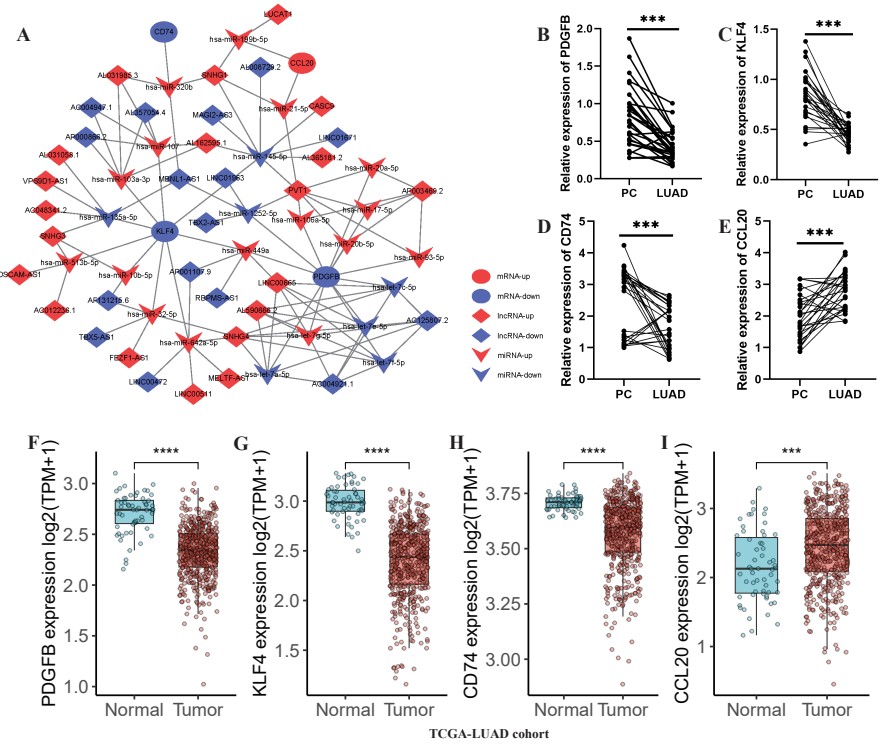

**Figure 8   Development of a TAM-associated lncRNA-miRNA-mRNA network in LUAD.** (A) A TAM-associated lncRNA-miRNA-mRNA network was constructed based on the 36 DE-lncRNAs, 23 shared-miRNA, and four TAMGs (PDGFB, CD74, KLF4, and CCL20). (B–E) QPCR validated the expression of PDGFB, CD74, KLF4, and CCL20 in LUAD tumor tissues compared with paracancerous non-tumor tissues ($n = 28$). (F–I) The expression of PDGFB, CD74, KLF4, and CCL20 in LUAD tumor tissues ($n = 486$) compared with non-tumor tissues ($n = 59$) in TCGA-LUAD cohort.

*et al., 2021*; *Modak et al., 2022*). Targeting TAMs represents a promising therapeutic approach, particularly for treating intractably cold tumors.

In the present study, we identified a total of 316 DE-lncRNAs, 162 DE-miRNAs, and 2,601 DEGs in LUAD. From these DEGs, we selected six prognostic tumor-associated genes (TAMGs), including KLF4, GAPDH, PDGFB, TIMP1, CD74, and CCL20, for further analysis. LUAD patients were classified into high- and low-risk groups based on the median risk score, which was calculated using the expression levels of prognostic TAMGs and their corresponding regression coefficients. High-risk scores were significantly associated with poor survival, older age, male, aggressive tumor, and metastasis. Moreover, we found that high-risk scores were linked to the activation of several canonical cancer pathways, including the cell cycle, NOTCH, p53, pathways in cancer, proteasome, hypoxia, PI3K/AKT/mTOR, reactive oxygen species, and Wnt/beta-catenin signaling pathways.

Krüppel-like factor 4 (KLF4), also named as epithelial zinc finger protein (EZF) and gut-enriched Krüppel-like factor (GKLF), is a member of the evolutionarily conserved family of zinc finger transcription factors (*Garrett-Sinha et al., 1996*; *He, He & Xie, 2023*; *Shields, Christy & Yang, 1996*). KLF4 may function as an oncogene in NSCLC that is

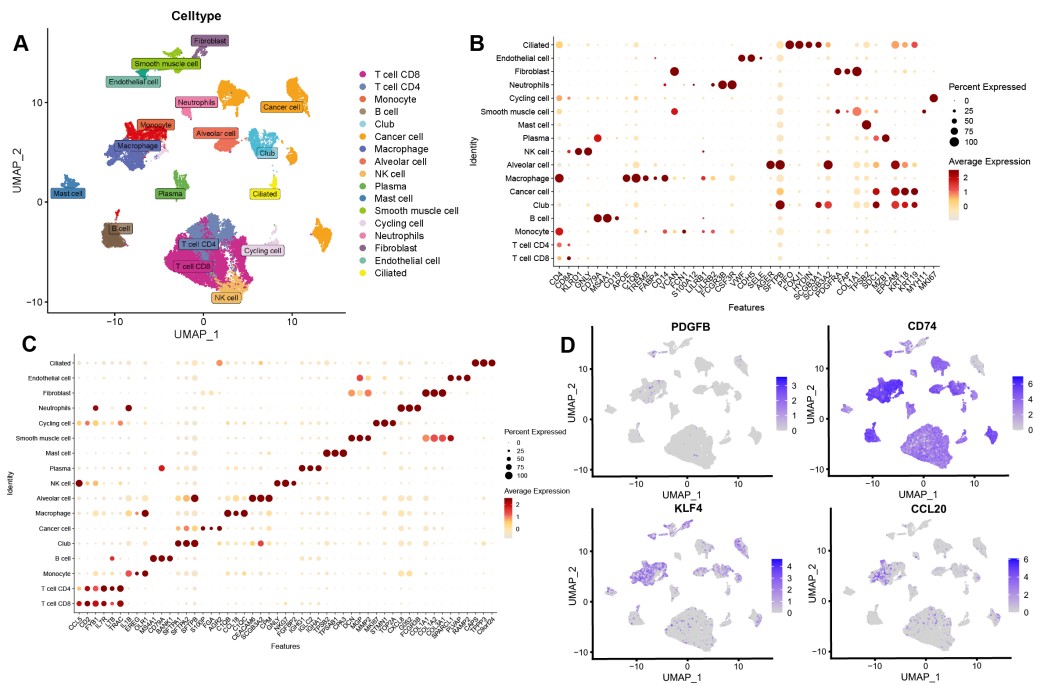

**Figure 9 Validation of the key TAMGs expression in monocyte/macrophage at the single-cell level.** (A) Cells were clustered into 17 types *via* UMAP, each color represented the annotated phenotype of each cluster. (B–C) Bubble plots of the identified marker genes and top three marker genes of LUAD in different cell types. (D) Feature plots of PDGFB, CD74, KLF4, and CCL20 expression in different cell types.

involved in macrophage infiltration and polarization (*Arora et al., 2021*; *Zhou et al., 2022*). Glyceraldehyde-3-phosphate dehydrogenase (GAPDH) is an internal reference gene to quantitate DNA, RNA, and proteins in usual biological experiments (*Wisnieski et al., 2013*; *Zhang et al., 2015*). Recent studies have indicated that GAPDH is involved in aging and cellular senescence (*Guan, Crasta & Maier, 2022*; *Yang et al., 2021*), as well a its role as an oncogene in various tumors, correlating with immune infiltration (*Butera et al., 2019*; *Shen, Li & Wang, 2023*). Bioinformatics analyses have revealed that GAPDH plays a senescence-related marker in LUAD, influencing cancer progression (*Liu et al., 2023*).

Overexpression of platelet-derived growth factor B (PDGFB) has been observed in several solid tumors, including pancreatic cancer, gastric cancer, glioma, melanoma, renal carcinoma, and breast cancer (*Abuhamad et al., 2023*; *Du et al., 2021*; *Juliano et al., 2018*; *Kadrmas, Beckerle & Yoshigi, 2020*; *Wang et al., 2021*). In this study, we identified high PDGFB expression as associated with TAM-related high-risk scores.

Tissue inhibitor of metalloproteinase-1 (TIMP-1) is a member of the inhibitor metalloproteinase (TIMP) family, traditionally considered a potential tumor suppressor. However, overexpression of TIMP has been linked to poor survival in lung cancer (*Duch et al., 2022*; *Fong et al., 1996*). In the present study, we also demonstrated that high expression of TIMP involves poor survival of LUAD. CD74, a type II transmembrane protein, acts as a receptor for the cytokine macrophage migration inhibitory factor (MIF). Upon binding MIF, CD74 releases its cytosolic intracellular domain (CD74-ICD), which serves as a

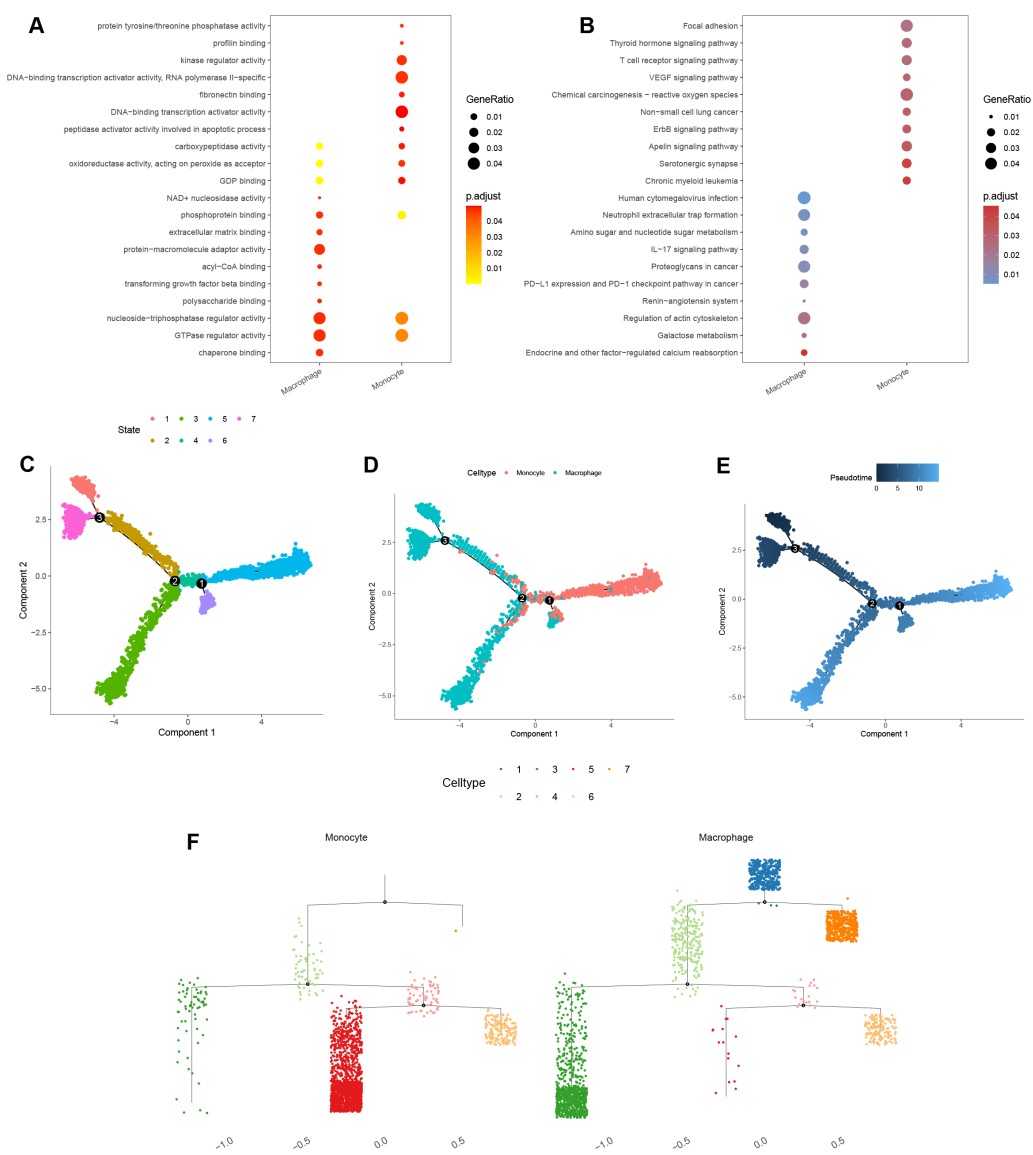

**Figure 10  Functional and pseudotime trajectory analyses of the key TAMGs expression in monocyte/macrophage at the single-cell level.** (A–B) Bubble plots of the GO and KEGG pathway enrichment in monocyte and macrophage cell lineages. (C–E) Scatter plot showing the pseudotime trajectory in the monocyte/macrophage lineages based on cell state, cell types, and pseudotime score, respectively. (F) Trajectory tree diagram of the pseudotime trajectory in the monocyte/macrophage lineages.

transcriptional regulator in normal B cells (*David et al., 2022*). CD74 is associated with favorable prognosis in patients with HCC (*Xiao et al., 2022*). In contrast, we found that high CD74 expression in LUAD was linked to favorable survival outcome.

Chemokine (C-C motif) ligand 20 (CCL20), also known as macrophage inflammatory protein (MIP)-3α, liver activation-regulated chemokine (LARC), and Exodus-1, is a small protein that binds to the specific receptor C-C chemokine receptor 6 (CCR6) (*Kadomoto, Izumi & Mizokami, 2020*). Elevated CCL20 expression has been associated with poor

survival in patients with LUAD (*Fan et al., 2022*). Additionally, CCL20 promotes lung cancer cell migration and proliferation in an autocrine manner through the activation of ERK and PI3K signal pathways (*Wang et al., 2016*).

In addition, we explored the correlation between risk score and tumor immunity. We found high-risk scores were significantly associated with lower Immune score and ESTIMATE score, as well as an increased faction of macrophages (M0), T cells CD4 memory activated, neutrophils, NK cells resting, Mast cells activated. Conversely, high-risk scores were associated with decreased the fraction of T cells CD4 memory resting, mast cells resting, dendritic cells resting, Tregs, monocytes, and B cells memory.

Furthermore, high-risk scores were correlated with low expression of HLA-related genes and immune checkpoints (CD80, CTLA4, and TIGIT), higher TIDE scores, and lower IPS scores in LUAD. These findings suggest that high-risk scores may be linked to immune evasion and therapeutic resistance. TAMs are recognized as potential therapeutic targets due to their modifiable polarization states within the TME (*Hu et al., 2022*; *Liang et al., 2022*; *Xu et al., 2020*; *Zhang et al., 2020*). Our finding highlights the role of TAMs in modulating tumor progression by either promoting or inhibiting tumorigenesis within the TME.

Finally, we explored the potential regulatory mechanisms of prognostic TAMGs in LUAD. A lncRNA-miRNA-mRNA network was constructed, containing of DE-lncRNAs, 23 shared-miRNA, and four TAMGs (PDGFB, CD74, KLF4, and CCL20). In the present study, several lncRNAs, such as lncRNA SNHG1 (*Li & Zheng, 2020*), lncRNA SNHG3 (*Li et al., 2021*), lncRNA SNHG4 (*Wang & Quan, 2021*), lncRNA CASC9 (*Bing et al., 2021*), and lncRNA PVT1 (*Pan et al., 2020a*), were identified as oncogenes, showing significant upregulation in lung cancer and involvement in tumorigenesis and development.

At the single-cell level, a total of 17 cell types were annotated in LUAD, with CD74, KLF4, and CCL20 significantly upregulated, while PDGFB was downregulated in monocyte/macrophages. Cell trajectory and pseudo-time analyses of monocyte and macrophage cells revealed a transition from monocytes to macrophages, consistent with the understanding that resident macrophages originate from hematopoietic stem cells and monocyte-derived macrophages (*Lazarov et al., 2023*).

## CONCLUSION

In conclusion, we conducted an integrative analysis of tumor-associated macrophages (TAMs) in LUAD, highlighting their association with poor survival, immune cell infiltration, and responses to immunotherapy and chemotherapy. Additionally, we explored the regulatory mechanisms of TAMGs and lncRNAs. Our findings provide potential therapeutic targets and biomarkers for patients with LUAD, offering valuable insights for future clinical applications.

### Funding

The authors received no funding for this work.

### Competing Interests

The authors declare there are no competing interests.

### Author Contributions

- Chaoqun Yu performed the experiments, authored or reviewed drafts of the article, and approved the final draft.
- Jun Chen performed the experiments, authored or reviewed drafts of the article, and approved the final draft.
- Jianming Deng performed the experiments, authored or reviewed drafts of the article, and approved the final draft.
- Hui Li conceived and designed the experiments, analyzed the data, prepared figures and/or tables, and approved the final draft.
- Qianru Zhuang analyzed the data, prepared figures and/or tables, and approved the final draft.
- Bingqing Luo analyzed the data, prepared figures and/or tables, and approved the final draft.
- Hua Ye conceived and designed the experiments, authored or reviewed drafts of the article, and approved the final draft.
- Hui Tian conceived and designed the experiments, authored or reviewed drafts of the article, and approved the final draft.

### Human Ethics

The following information was supplied relating to ethical approvals (*i.e.*, approving body and any reference numbers):

This study has been approved by the Ethical Committee of Yueqing People's Hospital (YQYY202300221).

### Data Availability

Raw data is available in the Supplemental Files.

### Supplemental Information

Supplemental information for this article can be found online at http://dx.doi.org/10.7717/peerj.19920#supplemental-information.

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
