# Peer review of "Integrative analyses of single-cell and bulk RNA sequencing to construct the tumor-associated macrophage-related prognostic signature in lung adenocarcinoma"

_PeerJ, doi:10.7717/peerj.19920_

## Round 0.1 · original submission · Major Revisions

Dear authors,

We kindly request that you carefully review the comments provided by the reviewers. Their valuable suggestions offer insights to enhance your manuscript. Incorporate their suggestions and carefully address all comments in your manuscript; it will significantly strengthen its content.

Reviewer 1 ·

Basic reporting

Yu et al integrate published single cell and bulk transcriptomic data to identify molecular signatures for predicting survival outcomes in lung adenocarcinoma. They specifically focus on tumor associated macrophages for their analyses and identify differentially expressed genes and RNA species that can be used for prognostic signatures. The manuscript needs a thorough revision of English and grammar. There are many grammatical errors and errors in sentence construction throughout the manuscript and I have difficulty in understanding few sentences.

Experimental design

The use of single cell data needs more rationale

Validity of the findings

The authors need to spell out clearly why they constructed the RNA networks in the manuscript. In addition, I have difficulty in understanding why single cell data was used. It appears that single cell data from lung tumors was used and without performing integration with a normal tissue single cell sample any conclusion cannot be reached. The authors present that genes CD74, KLF4, CCL20, PDGFB are specific to macrophages in tumors, but they may be also present in normal tissue samples. Also, CD74 and KLF4 are not specific to macrophages. There needs to be a thorough revision for this manuscript to be accepted.

Additional comments

- Line 32 in Background: No hypen between tumor and infiltrating
- Line 40: Capitalize differentially
- Line 67: please correct the sentence
- The manuscript has excessive use of many acronyms and is confusing at times. I suggest changing a few. For instance, its okay to write lung adenocarcinoma instead of LUAD. Other examples are TAMGs, TME etc
- I have provided a marked up file of the manuscript with some changes. Please refer.

Annotated reviews are not available for download in order to protect the identity of reviewers who chose to remain anonymous.

Reviewer 2 ·

Basic reporting

Tumor-associated macrophages (TAMs) play a role in tumor progression, but their roles in lung adenocarcinoma (LUAD) are not well understood. This study aimed to develop a TAM-related prognostic signature to predict survival outcomes and constructed a lncRNA-miRNA-mRNA network based on TAM-related genes. Transcriptomic, clinical, and single-cell RNA-sequencing data from TCGA and GEO databases were used to identify differentially expressed lncRNAs, miRNAs, and mRNAs in LUAD. The authors identified a TAM-related prognostic signature to predict the survival outcome of patients, and use qPCR to validated the expression of these genes.

There are some grammar issues and typos, some examples:
1. However, the roles of TAM and its regulatory mechanism in lung adenocarcinoma (LUAD) are limited.
2. We also constructed a lncRNA-miRNA-mRNA network was constructed based on the 36 DE-lncRNAs, 23 shared-miRNA, and 4 TAMGs (PDGFB, CD74, KLF4, and CCL20).
3. In the past two decades, advancements in targeted therapy and immunotherapy for patients with NSCLC that emerged in declines in lung cancer incidence.
4. the RAN-seq data of a total of 486 LUAD and 59 normal samples.
5. entire CGA-LUAD set

Experimental design

The authors mention that "Tumor-associated macrophages (TAMs) are a major population of tumor-infiltrating immune cells, playing dual roles in tumor progression with two main phenotypes: M1, which represents the tumor-suppressing subtype, and M2, which represents the tumor-promoting subtype." In Figure 5E, we observe changes in M0 but not in M1 and M2. What is the function of M0, and why is it associated with worse clinical outcomes?

The authors should plot the expression levels of PDGFB, KLF4, CD74, and CCL20 in TCGA LUAD and normal samples to compare them with Figure 8B.

Validity of the findings

The authors used independent datasets to test their hypothesis, ensuring robust conclusions.

·

Basic reporting

Extensive editing of sentence construction needed throughout the manuscript.

Sentence construction- 310-322, 378-379, 387-390

Experimental design

no comments

Validity of the findings

Tumor-associated macrophages (TAMs) are immune cells that infiltrate tumors and play critical roles in the progression of lung adenocarcinoma (LUAD). You et al have proposed a new prognostic signature related to TAMs to predict survival outcomes and aimed to .construct a network involving long non-coding RNAs (lncRNAs), microRNAs (miRNAs), and mRNAs based on these genes, further exploring their interactions in LUAD.
However, no experimental data has been presented to substantiate these conclusions.

The authors have validated the gene expression using qPCR using GAPDH
as the reference gene. However in their analysis the authors identified GAPDH as one of the DE genes. Thus, qPCRT analysis needs to be re-done with another reference gene.

They need to further extend this study to understand the expression of the miRNAs and lncRNAs from the constructed network targeting these genes. The protein expression of some of these genes should also be studied by histopathology in patient samples. Furthermore, there is no evidence to suggest that these genes affect the transition from monocytes to macrophages.

---

## Round 0.2 · Minor Revisions

Please address all of the issues raised by the reviewer.

Reviewer 2 ·

Basic reporting

I appreciate the authors' effort to address the grammar issues I pointed out.

Experimental design

They responded to my earlier question about the role of M0 in the study, but this explanation was not included in the revised manuscript.
I also suggested that they plot the expression levels of PDGFB, KLF4, CD74, and CCL20 in normal vs. tumor samples using the TCGA LUAD dataset. This suggestion was ignored without explanation. Additionally, I don't understand why Figures 8B–D were removed in the revised version.

Validity of the findings

no comment

·

Basic reporting

no additional comments

Experimental design

no additional comments

Validity of the findings

no additional comments

Additional comments

no additional comments

---

## Round 0.3 · Minor Revisions

Reviewer 2 ·

Basic reporting

I appreciate the authors' efforts in improving the manuscript; the revised version is significantly better.

Experimental design

The authors added the analysis for the TCGA dataset and confirm their QPCR results.

However, there are still a few minor issues with the figure captions that need addressing:
Figure 8: Descriptions for panels F-I are missing.
Figure 9: The current figure only displays panels A-D, but the caption erroneously includes descriptions for panels E-J.

Validity of the findings

No comment.

---

## Round 0.4 · accepted · Accept

The authors have now addressed all of the issues raised by the reviewers.

Reviewer 2 ·

Basic reporting

The authors addressed all previous issues.

Experimental design

no comment

Validity of the findings

no comment